# Study of the Female Sex Survival Advantage in Melanoma—A Focus on X-Linked Epigenetic Regulators and Immune Responses in Two Cohorts

**DOI:** 10.3390/cancers12082082

**Published:** 2020-07-28

**Authors:** Abdullah Al Emran, Jérémie Nsengimana, Gaya Punnia-Moorthy, Ulf Schmitz, Stuart J. Gallagher, Julia Newton-Bishop, Jessamy C. Tiffen, Peter Hersey

**Affiliations:** 1Melanoma Oncology and Immunology Program, The Centenary Institute, The University of Sydney, Royal Prince Alfred Hospital, Missenden Road, Camperdown NSW 2050, Australia; a.alemran@centenary.org.au (A.A.E.); g.punniamoorthy@centenary.org.au (G.P.-M.); s.gallagher@centenary.org.au (S.J.G.); j.tiffen@centenary.org.au (J.C.T.); 2Melanoma Institute Australia, The University of Sydney, Sydney NSW 2006, Australia; 3Leeds Institute of Medical Research at St James’s, University of Leeds, Leeds LS2 9JT, UK; J.Nsengimana@leeds.ac.uk (J.N.); J.A.Newton-Bishop@leeds.ac.uk (J.N.-B.); 4Biostatistics Research Group, Population Health Sciences Institute, Faculty of Medical Sciences, Newcastle University, Newcastle upon Tyne NE1 7RU, UK; 5Computational Biomedicine Lab Centenary Institute, The University of Sydney, Camperdown NSW 2050, Australia; u.schmitz@centenary.org.au; 6Gene & Stem Cell Therapy Program Centenary Institute, The University of Sydney, Camperdown NSW 2050, Australia; 7Faculty of Medicine and Health, The University of Sydney, Camperdown NSW 2050, Australia

**Keywords:** melanoma, sex difference, X-linked genes, X chromosome inactivation, epigenetic regulators, histone demethylase, EZH2 inhibitor, GSEA, immune response, survival, TCGA, SKCM, LMC

## Abstract

Background: Survival from melanoma is strongly related to patient sex, with females having a survival rate almost twice that of males. Many explanations have been proposed but have not withstood critical scrutiny. Prior analysis of different cancers with a sex bias has identified six X-linked genes that escape X chromosome inactivation in females and are, therefore, potentially involved in sex differences in survival. Four of the genes are well-known epigenetic regulators that are known to influence the expression of hundreds of other genes and signaling pathways in cancer. Methods: Survival and interaction analysis were performed on the skin cutaneous melanoma (SKCM) cohort in The Cancer Genome Atlas (TCGA), comparing high vs. low expression of *KDM6A*, *ATRX*, *KDM5C,* and *DDX3X*. The Leeds melanoma cohort (LMC) on 678 patients with primary melanoma was used as a validation cohort. Results: Analysis of TCGA data revealed that two of these genes—*KDM6A* and *ATRX*—were associated with improved survival from melanoma. Tumoral *KDM6A* was expressed at higher levels in females and was associated with inferred lymphoid infiltration into melanoma. Gene set analysis of high *KDM6A* showed strong associations with immune responses and downregulation of genes associated with Myc and other oncogenic pathways. The LMC analysis confirmed the prognostic significance of *KDM6A* and its interaction with *EZH2* but also revealed the expression of *KDM5C* and *DDX3X* to be prognostically significant. The analysis also confirmed a partial correlation of *KDM6A* with immune tumor infiltrates. Conclusion: When considered together, the results from these two series are consistent with the involvement of X-linked epigenetic regulators in the improved survival of females from melanoma. The identification of gene signatures associated with their expression presents insights into the development of new treatment initiatives but provides a basis for exploration in future studies.

## 1. Introduction

Melanoma is an aggressive form of skin cancer and is the most common type of cancer in young Australian adults. In Australia and most western countries, females have much better survival than male patients [1,2,3]. As an example, in 2012, the Australian age-adjusted death rate for females was approximately 2 per 100,000 of the population compared to 5 per 100,000 in males (www.aihw.gov.au). A number of reasons for the female survival advantage have been proposed, such as behavioral differences in sun exposure, leading to differences in incidence and stage at diagnosis [4], presentation of males with more advanced thicker melanoma [5,6], higher mutation rates in melanoma from males [7], and inhibition by estrogens in females (reviewed in [8]). Nevertheless, multivariate analyses in a number of studies have confirmed that the female sex is an independent prognostic indicator in survival from melanoma [1,9,10].

Improved survival of female patients has been noted in other malignancies and has prompted an examination of the role of X-linked genes that escape X chromosome inactivation. One such study from The Cancer Genome Atlas (TCGA) identified six X-linked genes that harbored loss of function mutations in cancers from males that, therefore, might play protective roles in females [11]. Three of these genes—*KDM5C*, *KDM6A,* and *ATRX*—are well-known epigenetic regulators with known tumor-suppressive functions in melanoma. *KDM5C* and *KDM6A* have been previously shown to be preferentially expressed in primary melanoma in female patients [12] and that the low levels of tumoral *ATRX* are associated with the progression of melanoma [13].

The identification of KDM6A (lysine(k) demethylase 6A) in this context is of particular interest as it is involved in the demethylation of Histone H3 Lysine 27 tri-methylation (H3K27me3) associated with gene activation. It is also known as UTX (ubiquitously transcribed X chromosome tetratricopeptide repeat protein) and is a member of the KDM6 family that includes UTY (Ubiquitously Transcribed Tetratricopeptide Repeat containing Y-linked) and KDM6B (also known as JMJD3, encoded by an autosomal gene) [14]. H3K27me3 is generated by the methylase EZH2(Enhancer of Zesta Homolog-2), which is the catalytic subunit of the polycomb-repressive complex 2 (PRC2) that represses transcription of genes involved in differentiation and tumor suppression in many cancers, including melanoma [15,16,17]. KDM6B also catalyzes the demethylation of H3K27me3, whereas UTY lacks demethylase activity due to the substitution of critical amino acids within the JmjC domain [18]. KDM6A is, therefore, a possible antagonist of PRC2 repression of these genes, initiating transcriptional activation. KDM5C is also a demethylase but demethylates histone H3K4. Methylation of H3K4 is considered to have a role in the activation of gene expression rather than repression [19].

ATRX (alpha thalassemia mental retardation X-linked) is a member of the SWI/SNF(SWItch/Sucrose Non-Fermentable) family of chromatin regulators, and its loss of function was first recognized as a cause of the ATRX syndrome. It has domains that can bind DNA methyltransferases, heterochromatin 1 alpha (HP1alpha), and EZH2 [20] and appears to have a role in suppression of repetitive regions of the genome like endogenous retroviral elements (ERV) and pericentric chromosome regions [20]. Its loss has also been associated with the maintenance of telomeres by the alternate lengthening of telomeres (ALT) pathways [20,21]. Most importantly, it is believed to be responsible for targeting the PRC2 complex to facilitate X chromosome inactivation and in targeting PRC2 to its target genes in the genome [22].

To identify X-linked genes that might be involved in the improved survival of females in melanoma, we selected genes that were associated with improved survival in females compared to males in the TCGA melanoma data. The expression of these genes was further examined for their association with immune infiltrates in primary and metastatic melanoma and for their association with particular gene sets associated with oncogenesis or immune-related pathways. The results provided strong evidence for the importance of *KDM6A* and *EZH2* in the improved survival of females from melanoma. An independent data set derived from 687 patients with primary melanoma in the Leeds melanoma cohort (LMC) [23] was used as the validation set.

## 2. Results

### 2.1. High KDM6A and ATRX Expression Are Associated with Improved Survival in Female Compared to Male Patients with Melanoma in the TCGA Data Set

The six pan-cancer X-linked genes (*ATRX*, *CNKSR2*, *DDX3X*, *KDM5C*, *KDM6A,* and *MAGEC3*) [11] were examined for their association with melanoma survival by comparing the survival of all patients with the expression of genes above (high) or below (low) median expression levels. Only *KDM6A* and *ATRX* were associated with improved survival. *KDM5C*, *DDX3X,* and *MAGEC3* were not associated with survival, and *CNKSR2* appeared to be associated with improved survival of males but not females (Figure 1a).

Kaplan–Meier analysis showed that the high *KDM6A* mRNA expression was associated with better overall survival in the skin cutaneous melanoma (SKCM) melanoma cohorts (*p* < 0.05, Figure 1b). As a control, we examined the expression of the paralog *KDM6B*, but it was not related to survival (Figure 1c). Women had higher mRNA expression of *KDM6A* than men (Figure 1d). The improvement in survival was confined to female patients if dichotomized above or below the median expression, as shown in Figure 1e,f. Interaction analysis did not reveal interactions of *KDM6A* with sex (*p* = 0.96) (Table 1a), so the improved survival in females is likely due to their higher levels of *KDM6A*. Only *EZH2* significantly interacted with sex (*p* = 0.02) (Table 1a), while *KDM6A* had a statistically significant interaction with *EZH2* (*p* = 0.03). Survival analysis was further segregated in primary and metastatic disease and sex in the SKCM cohort. We did not observe any significant association of *KDM6A, ATRX,* and *EZH2* in primary or metastatic cohorts, suggesting that the survival advantage applied to both disease states (Appendix A).

High expression of *ATRX* was also associated with improved survival in melanoma patients. (Figure 2a). To assess whether high expression of two X-linked genes—*KDM6A* and *ATRX*—have a survival advantage, we stratified melanoma patients based on the median expression of *KDM6A* and *ATRX*. High *ATRX* was associated with improved survival regardless of *KDM6A* expression level and vice versa (Figure 2b), showing that *ATRX* is an independent predictor of survival, which was confirmed by univariate and multivariate cox regression analysis (Figure 2c). *ATRX* levels, however, were not significantly different between males and females (Figure 2d). There was also no significant interaction with sex (*p* = 0.14), *KDM6A* levels (*p* = 0.98), and *EZH2* levels (*p* = 0.31) (Table 1a).

#### LMC Analysis (Validation Cohort)

Many of these observations were replicated in the LMC primaries. Notably, higher expression of *DDX3X* and *KDM6A* was associated with better melanoma-specific survival in all patients (only when analyzed on a continuous scale for the former, Appendix A). *KDM6A* was expressed more highly in females, but its prognostic effect was similar in both sexes (interaction *p* = 0.41, Appendix A). Additionally, *KDM5C* expression was associated with improved survival in this cohort, unlike in TCGA, but the effect of KDM6B was flat in both cohorts. The expression of *ATRX* was not prognostic in the full data nor in any sex subset (interaction *p* = 0.47, Appendix A).

The melanoma-specific survival (MSS) in the LMC data also showed a weak increase in survival in patients with high KDM6A and low ATRX levels compared to that in patients with low levels of both KDM6A and ATRX (Appendix A).

### 2.2. EZH2 Levels Impact Survival in High KDM6A Patients and Have Significant Associations with Sex-Biased Overall Survival (OS)

The histone methyltransferase EZH2 and demethylase KDM6A enzymes have opposing effects on the H3K27 mark, so we examined whether *EZH2* levels in the TCGA data set would be associated with survival. The survival of patients with high *KDM6A* and either high or low expression of *EZH2* is shown in Figure 3a. Patients with high *KDM6A* and low *EZH2* levels had improved overall survival compared to low *KDM6A* and high *EZH2* patients (log-rank *p* = 0.014). High expression of *KDM6A* with low expression of *EZH2* was significantly associated with better survival, especially in females (log-rank *p* = 0.023) but not male SKCM patients (Figure 3b,c). Additionally, gene interaction analysis of *EZH2* and *KDM6A* on the OS suggested that *EZH2* interacted with *KDM6A* and impacted on overall survival in the total SKCM cohort, as well as in patients with metastases, but not in the primary cohort (interaction *p* = 0.033, 0.042, and 0.32 respectively, Table 1a). Moreover, *EZH2* also interacted with sex and had a significant association with survival in patients with primary or metastatic melanoma and overall SKCM cohort (interaction *p* = 0.045, 0.049, 0.02 respectively, Table 1a). However, no significant statistical interaction was observed between *EZH2* and *ATRX* on OS (interaction *p* = 0.313). *EZH2* expression was relatively higher in male compared to female patients; however, this was not significant (Appendix A). *EZH2* partially correlated with *KDM6A* in SKCM; however, paired analysis suggested that a subset of patients with high *KDM6A* level had a significantly low level of *EZH2* and vice versa (Appendix A). We also examined the association of age with the high *KDM6A* and low *EZH2* group since it has been reported that the improved survival of female melanoma patients only applies to those below 60 years of age [10]. However, multivariate cox regression analysis of our data did not find a relation between age and survival in the high *KDM6A* and low *EZH2* group (Appendix A).

#### LMC Analysis (Validation Cohort)

These results were similar to those from the LMC cohort, in that patients with high *KDM6A* levels and low *EZH2* levels had significantly better survival than patients with low *KDM6A* and *EZH2* levels. It was also of interest that female patients with low *EZH2* levels had better MSS than male patients with low *EZH2* levels (Appendix A). See also Table 1b, which confirms significant interactions between *KDM6A* and *EZH2* and *EZH2* and sex, respectively, on MSS and supports the results of the SKCM cohort.

### 2.3. The Tissue Distribution of KDM6A and ATRX Is Different to that of KDM6B

The improved survival of women from melanoma has been reported to be confined to women below the age of 60 [10], which prompted us to examine whether hormonal influences are at play. Tissue-specific mRNA expression data were retrieved from the GTEx portal [24]. *KDM6A* expression was higher in several female organs, including ovary, uterus compared to other tissues. *KDM6A* expression was higher in lymphocytes from female compared to male patients (Appendix A). Similarly, *ATRX* expression was relatively higher in several female organs (Appendix A). In contrast, *KDM6B* had a basal level of expression in the female organs (Appendix A).

### 2.4. Gene Set Expression Analysis (GSEA) Has Revealed Immune Pathways to Be Positively Associated with KDM6A

To get a better understanding of how *KDM6A* may be influencing survival, a differential gene expression analysis of the *KDM6A* high vs. low group was carried out on the TCGA data set. In the KDM6A high group, HALLMARK ALLOGRAFT REJECTION, INFLAMMATORY RESPONSE, and INTERFERON-GAMMA RESPONSE were the top three positively enriched pathways (normalized enrichment score > 1.9, *p* = 0) (Figure 4a). The most negatively enriched gene sets were HALLMARK OXIDATIVE PHOSPHORYLATION, MYC TARGET V2, DNA REPAIR (normalized enrichment score < −2.2, *p* = 0) (Figure 4b). Additionally, other immune-related pathways like HALLMARK IL6 JAK STAT3 SIGNALING and INTERFERON ALPHA RESPONSE were positively enriched amongst the top 10 pathways (Figure 4c). Several metabolic and oncogenic pathways were negatively enriched, including HALLMARK OF REACTIVE OXYGEN SPECIES, GLYCOLYSIS, MTORC1 SIGNALING, CHOLESTEROL HOMEOSTASIS (Figure 4c).

Next, we asked whether the positive enrichment of immune-related pathways in the *KDM6A* group is associated with sex. Hence, we performed GSEA analysis of female *KDM6A* high vs. low group and male *KDM6A* high vs. low group. This analysis revealed a significant positive enrichment of HALLMARK INTERFERON-GAMMA RESPONSE in female patients with high *KDM6A* but not in male patients (Figure 5a). Immune-related pathways were absent in male patients. These results were consistent with *KDM6A*-related immune enrichment as the basis for survival advantage in females (Figure 5b). Both male and female patients shared some negative enriched pathways like HALLMARK OXIDATIVE PHOSPHORYLATION, DNA REPAIR, and GLYCOLYSIS, but EPITHELIAL MESENCHYMAL and MTORC1 SIGNALING were confined to female patients (Figure 5a,b).

### 2.5. GSEA Analysis Shows that ATRX Acts on Similar Gene Sets to KDM6A Except for Immune Regulatory Pathways

Next, we examined the TCGA data by GSEA to see which gene sets were associated with *ATRX*. As shown in Figure 6a–c, *ATRX* was associated with the upregulation of similar pathways to KDM6A, except for the immune pathways shown in Figure 4c. HALLMARK OXIDATIVE PHOSPHORYLATION, MYC TARGET V2, and DNA REPAIR pathways were identified as the top three negatively enriched pathways (normalized enrichment score < −2.2, *p* = 0) similar to the *KDM6A* analysis (Figure 6b,c).

### 2.6. GSEA Analysis Identifies a Positive Enrichment of Immune Pathways in KDM6A High EZH2 Low Group Similar to KDM6A High Group

As EZH2 and KDM6A act in an opposing fashion on H3K27 methylation, we compared tumors with high *EZH2* and low *KDM6A* levels to those with low *EZH2* and high *KDM6A* levels. These groups were stratified based on the median gene expression of *KDM6A* and *EZH2* in SKCM cohorts. GSEA revealed several immune-related pathways like HALLMARK INFLAMMATORY RESPONSE, INTERFERON-GAMMA RESPONSE, COMPLEMENT, and INTERFERON ALPHA RESPONSE amongst the top 10 positively enriched pathways in *KDM6A* high-*EZH2* low group (normalized enrichment score > 1.9, *p* = 0) (Figure 7a,b). On the other hand, HALLMARK E2F TARGETS, G2M CHECKPOINT, MYC TARGETS, OXIDATIVE PHOSPHORYLATION, DNA REPAIR, MTORC1 SIGNALING identified as one of the top 10 negatively enriched pathways (normalized enrichment score < −2, *p* = 0) (Figure 7b,c). These results were consistent with the improved survival due to KDM6A driven activation of immune-related pathways and modulation of the metabolic, cell cycle, and oncogenic pathways.

### 2.7. Association of KDM6A and ATRX on Immune Responses in Melanoma Patients

The original report on the genomic classification of cutaneous melanoma drew attention to the strong influence of lymphocytic infiltration on survival outcomes [25]. In view of this and the GSEA analysis showing that immune-related pathways were positively enriched in the patients with high *KDM6A,* we examined whether there was a correlation between *KDM6A* expression and immune infiltration scores, as reported previously [26]. This showed that *KDM6A* expression had significant positive correlations with several immune cell infiltrates, including dendritic cells (DCs) (r = 0.16), macrophages (r = 0.12), neutrophils (r = 0.31), B cells (r = 0.18), CD4+ (r = 0.22), and CD8+ cells (r = 0.32) (Table 2). Importantly, this association was more prominent in female patients compared to males.

*ATRX* expression showed less positive correlations with dendritic cells (r = 0.11), macrophages (r = 0.17), neutrophils (r = 0.29), B cells (r = 0.12), CD4+ (r = 0.12), and CD8+ cells (r = 0.18). *ATRX* expression like that of *KDM6A* also showed a higher positive correlation with immune cell subsets in female patients compared to male. *KDM6B* showed only a positive correlation with CD4+ T cells (r = 0.15) and a negative correlation with CD8+ T cells (r = −0.18). There were no correlations with other subsets. *EZH2* expression showed a significant positive correlation only with CD8+ T cells (r = 0.18) and neutrophils (r = 0.18) (Table 2). Similar results were observed in the metastases cohort, and the correlation was stronger, particularly in female patients. These results inferred that *KDM6A*, more so than *ATRX*, plays an important role in immune infiltration in melanoma patients, especially in females.

#### LMC Analysis (Validation Cohort)

Comparisons of tumor-infiltrating lymphocytes (TILs) between the two series is difficult because of different scoring systems, but nevertheless, as shown in Appendix A, a similar analysis on the LMC data showed weak associations of *KDM6A* with cytotoxic cells, B cells, DCs, neutrophils, and macrophages. These associations were only evident in females, consistent with the TCGA data analysis. *ATRX* levels did not appear to correlate with any immune infiltrates (Appendix A).

### 2.8. Exploration of the Basis for Increased TILs in Females Compared to Males

To determine if particular immune subsets are associated with *KDM6A* in melanoma, we compared the expression of the markers in females versus males in the TCGA data. As shown in Table 3, there did not appear to be major differences in the presence of CD141 DCs [27], BATF3(Basic Leucine Zipper ATF-Like Transcription Factor 3) and CD103 DCs [28], Regulatory T cells (Tregs, CD25), T resident memory T cells (Trem, CD69) [29], or CXCR3 (C-X-C Motif Chemokine Receptor 3) CD4 T cells between the sexes, but there were significantly higher levels of IFN(Interferon) gamma and the chemokine receptor CCR5(C-C Motif Chemokine Receptor 5) in female patients.

In the LMC cohort, a weak correlation was observed with *CD141*, *CD69,* and *CXCR3* in females compared to males. Whereas, *CCR5* was significantly correlated with *KDM6A* in males compared to females (Appendix A).

## 3. Discussion

Studies in a number of western countries have shown major differences in survival from melanoma between the sexes that even surpass the improvement in survival from new treatments with targeted therapies and immunotherapy [30]. Although X-linked genes have attracted attention in studies on autoimmune diseases [31], they have received little attention in studies on sex differences in cancer. The basis of the present study was the report of X-linked genes in the improved survival of females with a range of cancers [11]. Our application of these findings to melanoma patients in the TCGA database appears to make a strong case for the involvement of two of these X-linked epigenetic regulators (*KDM6A* and *ATRX*) in this improved survival. Not only *KDM6A* and *ATRX* were significantly associated with better overall survival, but *KDM6A* was also particularly higher in female compared to male melanoma patients. *KDM6A* is located on the X-chromosome, but both alleles can be transcribed in females, giving higher expression levels [19]. This was reflected in higher levels in melanoma from female patients. These differences in survival were not seen in the autosomal *KDM6B* paralog of *KDM6A*.

In looking for an explanation for these findings, we first examined whether *KDM6A* might be related to immune responses to melanoma, particularly as the initial analysis of the TCGA data had shown that TILs were the main factor related to the survival [25]. It was of great interest to find that there was a much stronger association of *KDM6A* expression with TILs in melanoma in female compared to male patients. The validity of these findings was strengthened by the lack of association of immune infiltrates with *KDM6A* in male patients, and those similar associations did not apply to the expression of *KDM6B*—the autosomal paralog of *KDM6A*. Further strong support for an immune basis for these findings came from GSEA, comparing high with low levels of *KDM6A* that showed strong upregulation of genes in the interferon (IFN) and inflammatory pathways. Most importantly, GSEA analysis of *KDM6A* based on sex revealed that the IFN gamma pathway expression was confined to female patients and completely absent in male patients. This was consistent with the preferential survival advantage in female melanoma patients with high *KDM6A* levels. Effects on oncogenic pathways might have also been involved as there was a strong downregulation of Myc. As reviewed elsewhere, Myc has been implicated in the inhibition of T cell activation and infiltration [32], and its downregulation may have contributed to the increase in TILs seen in the *KDM6A* high female patients.

As strong as these results from the TCGA analysis appear, the comparison with results from the LMC with primary melanoma raises several questions. In particular, why there was not a strong relation of *KDM6A* expression to MSS in females compared to males, even though *KDM6A* levels were much higher in females. There was only a weak association between *KDM6A* levels and TILs, but this was consistent with TCGA results, being seen only in females. Multivariate analysis also did not support that *KDM6A* or *ATRX* was independent of other known prognostic variables. The main difference between these cohorts is that the TCGA data is heavily weighted towards metastases from skin and lymph nodes. The primary melanoma in the TCGA is also considered to be weighted to more advanced melanoma as tissue from early melanoma is likely to be in short supply. Other differences may exist in the proportion of male/female patients at different ages, as this does vary by country of origin [33]. For example, as expected, there were more females with melanoma at a younger age and a site on their legs in the LMC rather than at head and neck sites as in older males in the US and Australia and seen also in the TCGA data (Appendix A).

Several of these X-linked epigenetic regulators have been implicated in immune responses. KDM6A has been reported to have roles in the function of Treg cells [34,35] and CD4 T cells in autoimmune disease [35] and maintenance of Th17 (T-helper cell 17) CD4 T cell responses [36]. Considering the known association of autoimmunity to allograft rejection [37], this may explain the positive enrichment of ALLOGRAFT REJECTION in the *KDM6A* high group. It is implicated in T follicular helper cell-dependent clearance of viral infections [38] and in the development of inflammatory responses to respiratory syncytial virus infections [39]. At a molecular level, KDM6A is known to have an opposing role to EZH2 in methylation of H3K27 histones. This role may explain some of the effects of KDM6A on the immune system in that we previously reported that EZH2 was associated with the repression of several genes associated with antigen presentation and chemokines involved in T cell responses [40]. The opposing effect of KDM6A on EZH2 might explain the positive enrichment of immune-related genes in the *KDM6A* high and low *EZH2* group in the GSEA analyses. Along this line, our recent study identified immune-suppressive gene signatures in a subset of melanoma cell lines resistant to EZH2 inhibitors. Sensitivity to the EZH2 inhibitor is strongly associated with interferon-gamma and alpha gene signatures [40]. Consistently, we also observed a positive enrichment of INTERFERON-GAMMA and ALPHA RESPONSE in the *KDM6A* high group as well as *KDM6A* high-*EZH2* low group. This suggested that the level of EZH2 and its demethylase KDM6A may play a pivotal role in regulating genes associated with immune evasion in melanoma. It was noticeable that immune-regulated genes were absent in melanoma with high ATRX levels, which may indicate retargeting of EZH2 to immune gene sets.

The LMC analysis also showed a strong relation of *KDM5C* expression to survival from melanoma. Its expression in melanoma has been shown in previous studies to be sex-related [12]. The relation to survival is somewhat surprising as KDM5C demethylase acts on H3K4 methylated histone marks, which are viewed as activating marks in euchromatin. KDM5C has also been described as inhibiting the endogenous IFN pathway needed to generate immune responses [41]. It would be plausible if KDM5C downregulation occurred as a response to metastases as the LMC analyses were on primaries. DDX3X was weakly associated with survival in both series when examined as a dichotomous variable but strongly associated with survival in the LMC when analyzed as a continuous variable. DDX3X is an x-linked RNA helicase, which has been linked to post-translational regulation of MITF (Microphthalmia-associated Transcription Factor) expression and development of melanoma metastasis and therapy resistance [42]. It is possible that the female advantage in relation to DDX3X may result from the adverse effect of mutation or loss of DDX3X in male patients rather than effects on immune responses. These differences in the expression between the two-patient series require further study, particularly in relation to the stage of the disease.

Given that the TCGA analysis suggested a strong role for *ATRX* in the survival of patients, it is surprising that it seems of little significance in the survival of patients in the LMC. Low levels have been previously reported to be associated with melanoma progression [13]. It is a member of the SWI/SNF chromatin remodeling family and is believed to have a role in the maintenance of heterochromatin and repression of genes, as well as in telomere maintenance [20]. It is of particular interest for targeting the PRC2/EZH2 complex to facilitate X chromosome inactivation and for genome-wide directing the PRC2/EZH2 to target genes [22]. This role for ATRX could presumably account for the upregulation of EZH2 target genes in melanoma and the increased expression of the X chromosome-related genes, such as KDM6A.

Another unanswered question is whether the effects of female hormones on the immune system may play some role in the improved survival [43], particularly as some series reported less sex-related effects in women over 60 years of age. It is also notable that *KDM6A* expression (but not *KDM6B*) is higher in female endocrine tissues and lymphoid tissue [8,43,44]. It is generally recognized that females have higher CD4 T cell levels and lower Treg cells [45]. Treatment initiatives based on hormone administration might be one development if confirmed. The present results also suggested that EZH2 inhibitors might have a role in the treatment of melanoma that expresses high *EZH2* and low *KDM6A* as loss of KDM6A sensitizes bladder and lung cancer cells to treatment with EZH2 inhibitors [19]. The involvement of ATRX in directing EZH2 to its targets also points to further exploration of EZH2 inhibitors in the treatment.

## 4. Materials and Methods

### 4.1. Data from Skin Cutaneous Melanoma (SKCM) Cohort

The SKCM patient cohort was obtained from TCGA and fetched through the Oncolnc website (www.oncolnc.org), consisting of 458 patients who all had cutaneous melanoma [46,47]. The clinical information, RNA-seq expression profile, and immune infiltration score of the SKCM cohort were obtained from TCGA. The overall survival was defined as time to death. Written informed consent of all participants was obtained by the TCGA Research Network. All research activities were conducted in accordance with the Helsinki Declaration of 1975 by the TCGA Research Network [47]. The description of the patients in the TCGA and LMC are summarized in Appendix A.

### 4.2. RNA-seq Analysis

RNA expression profiles (RNA Seq V2 RSEM) and clinical data of 472 melanoma patient samples from the TCGA Firehose Legacy cohort were retrieved using the Cancer Genomics Data Server R package (cgdsr, v.1.2.10; github.com/cBioPortal/cgdsr). For pairwise comparisons of gene expression profiles, the patient samples were grouped into four cohorts based on the median expression of *KDM6A*, *KDM6A* within female and male, *ATRX, and EZH2.* The four cohorts are represented as (a) *KDM6A* high vs. *KDM6A* low, (b) *KDM6A* high vs. *KDM6A* low in females and *KDM6A* high vs. *KDM6A* low in males, (c) *ATRX* high vs. *ATRX* low, (d) *KDM6A* high and *EZH 2* low vs. *KDM6A* low and *EZH2* high.

Differentially expressed genes were determined using the Wald test, followed by multiple testing correction (Benjamini Hochberg) within the DESeq2 R package [48]. Genes with an absolute log2 fold change ≥ 2 and adjusted *p*-value ≤ 0.05 were considered significantly differentially expressed and used for further analyses.

### 4.3. Gene Set Enrichment Analysis (GSEA)

RNA-seq analysis of the SKCM cohort revealed differentially expressed genes in (a) *KDM6A* high vs. low group, (b) *KDM6A* high vs. low in female and male groups, (c) *ATRX* high vs. low group, and (d) *KDM6A* high-*EZH2* low vs. *KDM6A* low-groups. The differentially expressed genes from each ‘high’ vs. ‘low’ expression group were further analyzed for enriched pathways by GSEA tools of the Broad Institute [49]. Briefly, a pre-ranked differential gene expression list from each group was uploaded in GSEA tools and run for Hallmark gene sets. The positive and negative enriched pathways with a cut off false discovery rate (FDR) *p* < 0.05 were considered for significant pathways. The top ten positively and negatively enriched pathways for each group were then represented as a forest plot based on their normalized enrichment score.

### 4.4. Statistical Methods

TCGA statistical analysis was performed using SPSS Statistical software of IBM (ibm.com/au-en/analytics/spss-statistics-software) and Graphpad Prism 7 (graphpad.com/scientific-software/prism/). Univariate and multivariate cox proportional hazard analysis was performed using SPSS software. The correlation between gene expression and individual immune cell subset infiltration was assessed by Pearson r and two-tailed t-tests. Survival analysis was performed in Prism using Log-rank (Mantel–Cox) to assess the significance between the groups. Models were considered for testing the effect of the expression of each gene on the overall survival in the whole dataset as well as in stratified subsets by sex or by the level of another gene.

To ascertain differential prognostic effects of one gene by levels of another or by sex while avoiding spurious association that may arise from subsetting the data, a statistical interaction term was added to multivariable Cox-proportional hazards regression, and Kaplan–Meier curves were plotted jointly for both factors after dichotomizing gene expressions by the median.

### 4.5. Data from the Leeds Melanoma Cohort (LMC)

This population ascertained that the primary melanoma cohort was comprised of data and samples collected in the UK. The 0.6 mm cores were obtained from FFPE (Formalin Fixed Paraffin Embedded) specimens, where it was possible to obtain a sample using a tissue microarray needle, without destroying the block. LMC tumor transcriptome processing has been described previously (accession no. EGAS00001002922) [23,50]. RNA was extracted to generate whole-genome gene expression data (Illumina DASL HT12.4 array). Background correction and quantile normalization were applied; singular value decomposition was used to assess the confounding factors that were subsequently adjusted out. Participants in the LMC gave written informed consent; the study was conducted in accordance with international ethical guidelines (Declaration of Helsinki) and was approved by the national ethics committee (MREC 1/03/57 and PIAG3-09(d)/2003). This dataset (n = 697) was used to replicate the observations made in the TCGA cohort. Genes were tested for association with melanoma-specific survival (MSS) in univariate and multivariate Cox-proportional hazard models with and without statistical interactions. Kaplan–Meier curves were used to plot the results. Spearman correlation was calculated between key genes and immune cell scores inferred using the immunome approach, as previously reported [51]. These analyses were conducted within STATA v14 (StataCorp, College Station, TX, USA).

## 5. Conclusions

In conclusion, our analysis of melanoma data in TCGA and LMC cohorts is consistent with a role for these four X-linked epigenetic regulators in the improved survival of females compared to male patients with melanoma. Differences between the two series may reflect different stages of melanoma and sites of occurrence. Comparisons with other cohorts are needed, as well as mechanistic studies in vitro, to better understand the biology of these genes in melanoma. Further analysis is also needed to see whether female hormones influence their effects on the immune system and account for the reduced sex differences in older female patients. Given the huge differences in survival from melanoma according to sex, further studies on relation to immune responses and oncogenic pathways appear warranted.

## Figures and Tables

**Figure 1 cancers-12-02082-f001:**
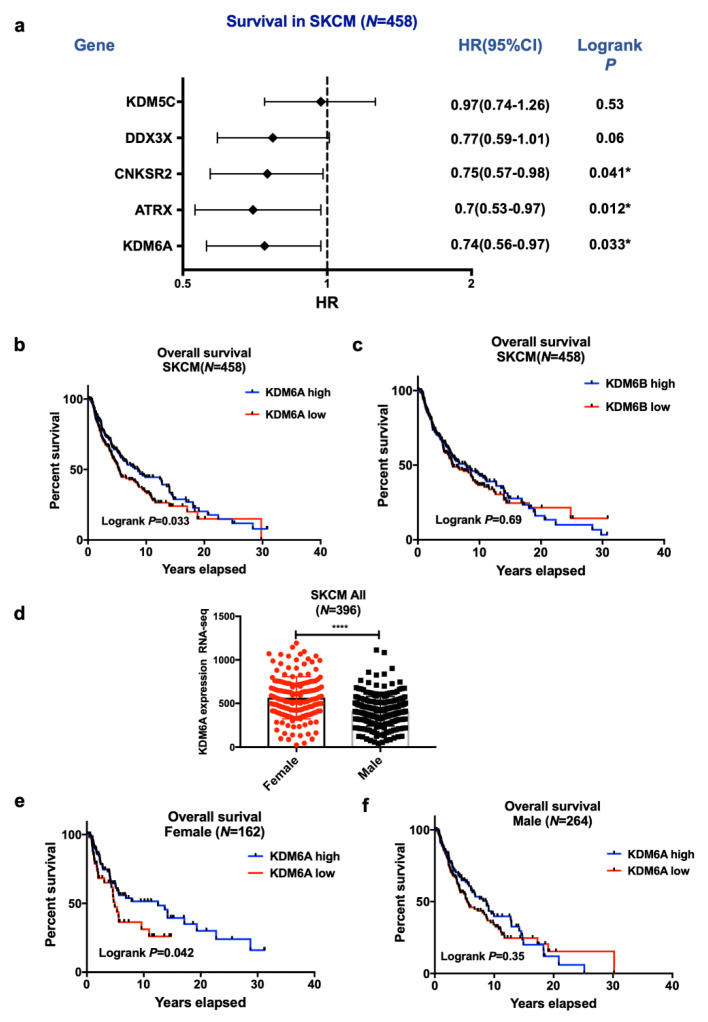
Influence of X-linked genes on skin cutaneous melanoma (SKCM) patient survival. (**a**) Forrest plot of hazard ratios for median survival according to high or low levels of the 5 X-linked genes. Low groups used as a baseline. The majority of the SKCM patients had no *MAGEC3* expression and were, therefore, excluded from the analysis. (**b**) A Kaplan–Meier (KM) plot showing overall survival of *KDM6A* high vs. low group and (**c**) *KDM6B* high vs. low group in SKCM patients. RNA-seq data were retrieved from The Cancer Genome Atlas (TCGA) and stratified based on the median expression of *KDM6A* and *KDM6B*. Log-rank *p*-value < 0.05 refers to a significant association between gene expression and survival. Differential gene expression and survival of *KDM6A* based on sex: (**d**) *KDM6A* mRNA expression data were plotted based on sex. An unpaired t-test was done to assess the significance between the two sexes, and the *p*-value is represented as (*), where **** *p* < 0.0001. (**e**) The overall curve was plotted in GraphPad Prism for female and (**f**) male SKCM patients. Log-rank *p* < 0.05 refers to significance. Gene expression was dichotomized based on sex and corresponding median expression.

**Figure 2 cancers-12-02082-f002:**
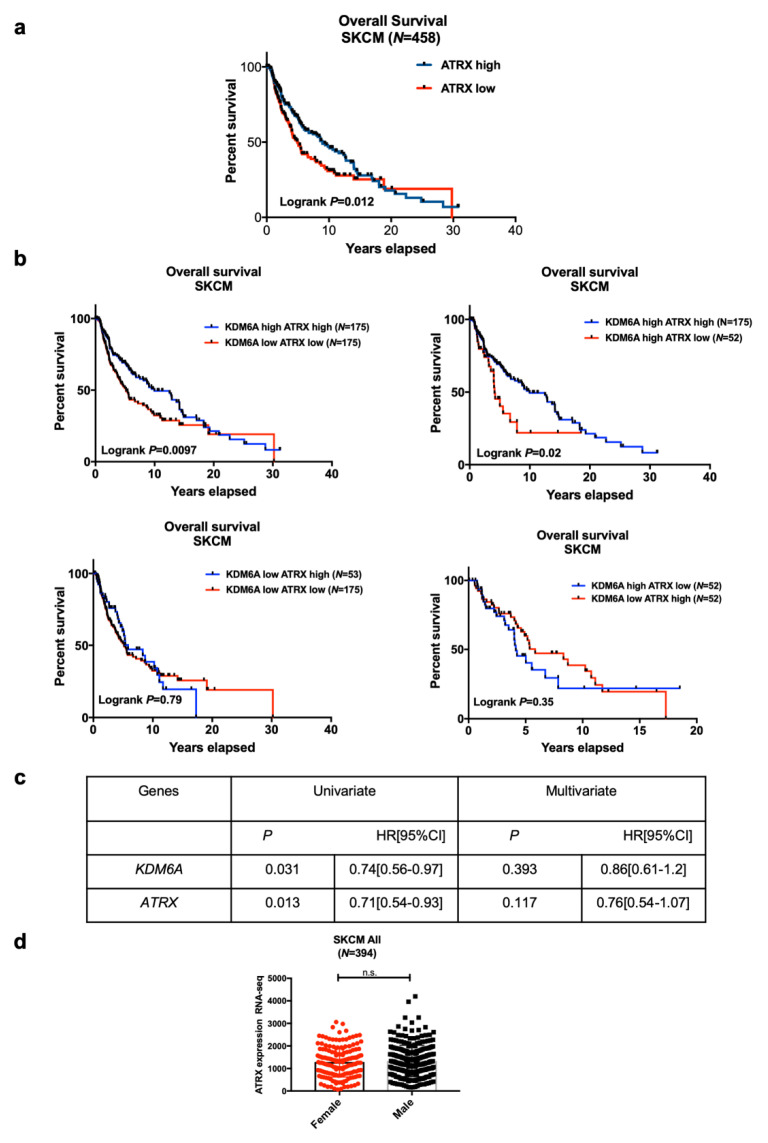
The analysis of patient survival according to the levels of *ATRX* and *KDM6A*. (**a**) A KM plot of overall survival was generated based on the median expression of alpha thalassemia mental retardation X-linked (ATRX). (**b**) SKCM mRNA expression data were stratified based on the median expression of KDM6A (lysine demethylase 6A) and ATRX, and the KM plot developed on *KDM6A-ATRX* high vs. *KDM6A-ATRX* low group, *KDM6A* high *ATRX* high vs. *KDM6A* high *ATRX* low group, *KDM6A* low *ATRX* high vs. *KDM6A* low *ATRX* high group, and *KDM6A* high *ATRX* low vs. *KDM6A* low *ATRX* high group. Log-rank *p* < 0.05 refers to a significant association. (**c**) Univariate and multivariate cox regression analyses were performed based on *KDM6A* and *ATRX* expression. *p* < 0.05 refers to significance. (**d**) *ATRX* expression based on sex in the SKCM cohort. Statistical analysis was performed by an unpaired *t*-test.

**Figure 3 cancers-12-02082-f003:**
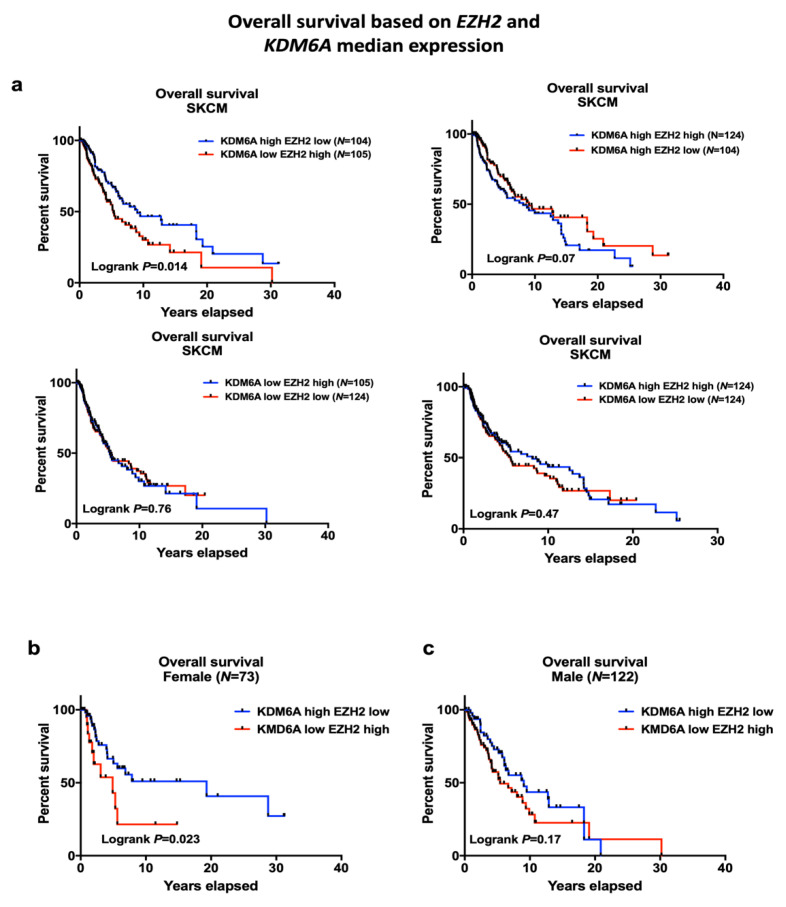
Overall survival of *KDM6A* high-*EZH2* low vs. *KDM6A* low-*EZH2* group. (**a**) A KM plot, showing overall survival of *KDM6A* high *EZH2* low vs. *KDM6A* low *EZH2* group, *KDM6A* high *EZH2* high vs. *KDM6A* high *EZH2* low group, *KDM6A* low *EZH2* high vs. *KDM6A* low *EZH2* low group, and *KDM6A* high *EZH2* high vs. *KDM6A* low *E2H2* low. SKCM patients were stratified based on the median expression of *KDM6A* and *EZH2*. (**b****,c**) Overall survival based on sex and *EZH2-KDM6A* median expression. *p* < 0.05 indicates the significance of survival.

**Figure 4 cancers-12-02082-f004:**
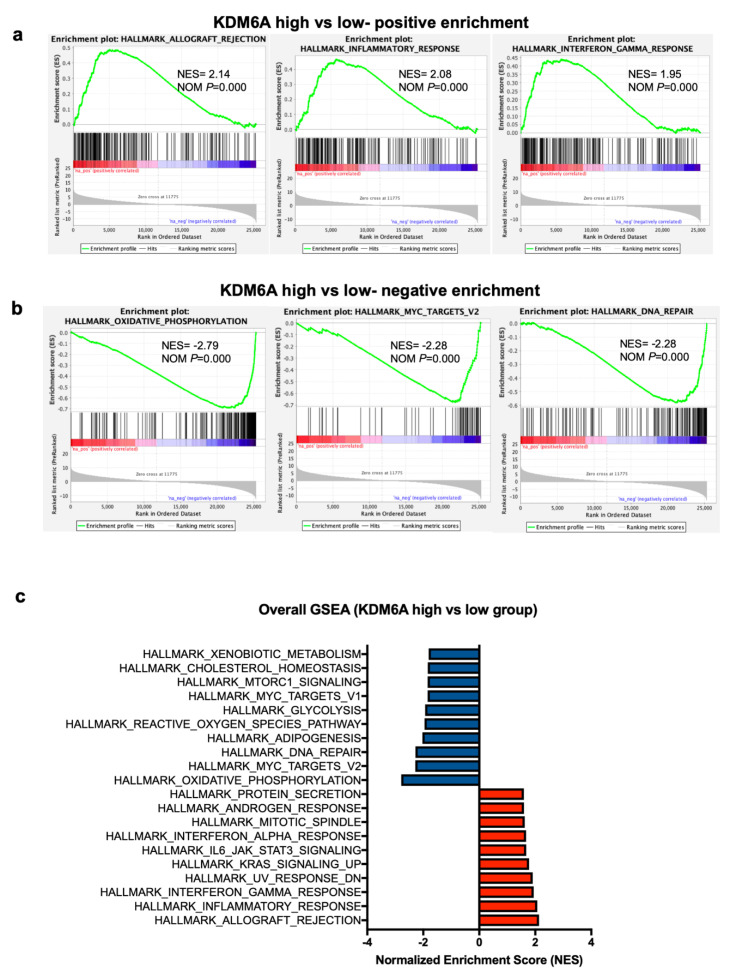
Gene set enrichment analysis (GSEA) of *KDM6A* high vs. low group. (**a**) Top 3 enrichment plots showing the pathways positively enriched in *KDM6A* high vs. low group. (**b**) Similarly, the top 3 enrichment plots, showing the pathways negatively enriched in KDM6A high vs. low group. (**c**) A forest plot was generated, showing overall GSEA pathways that are positively or negatively enriched according to the normalized enrichment score (NES). NOM, normalized *p*-value.

**Figure 5 cancers-12-02082-f005:**
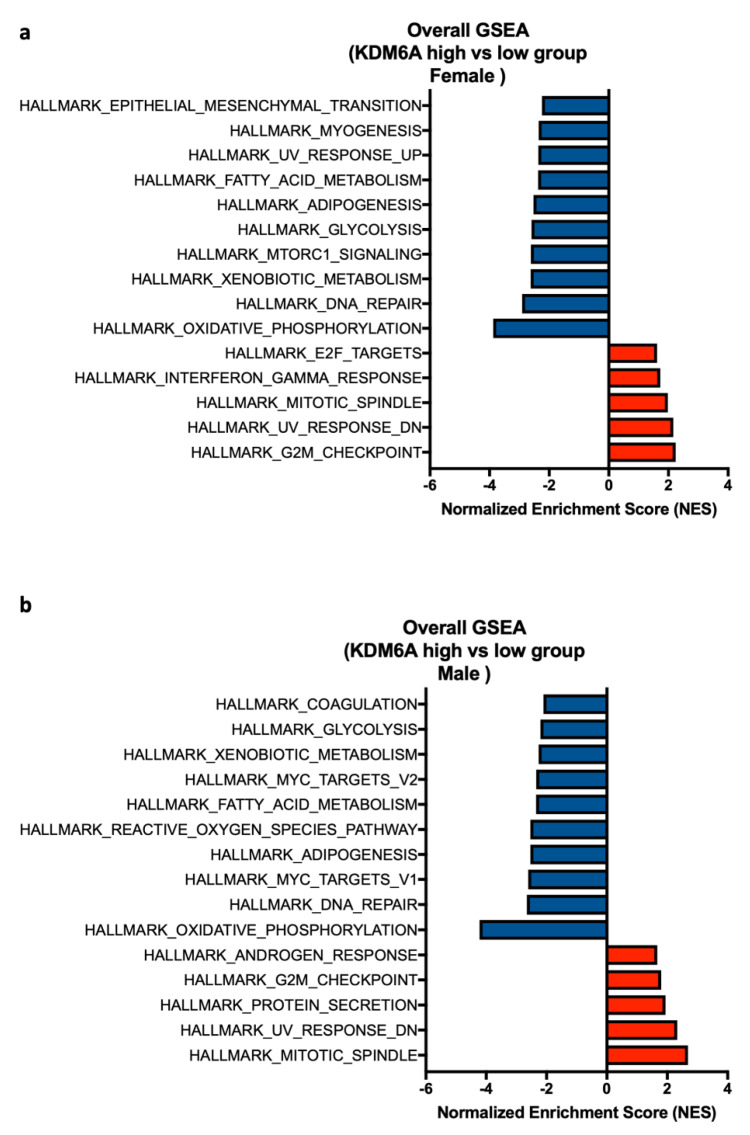
Gene set enrichment analysis of *KDM6A* high vs. low group based on sex. A forest plot represents the top 5 pathways that are positively enriched and 10 negatively enriched pathways using the NES score in (**a**) female and (**b**) male *KDM6A* high vs. low group. Pathways with normalized *p*-value < 0.05 refer to the significance and are shown in the plot.

**Figure 6 cancers-12-02082-f006:**
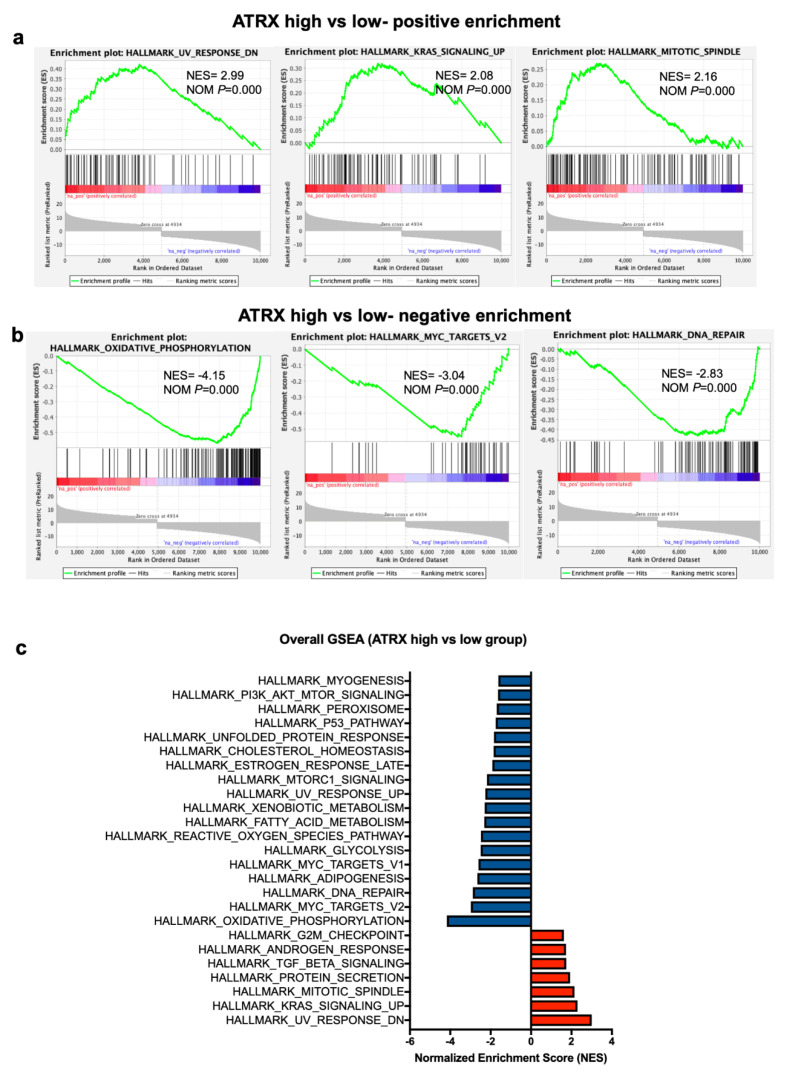
Gene set enrichment analysis of *ATRX* high vs. low group. (**a**) Top 3 enrichment plots showing the pathways positively and (**b**) negatively enriched in *ATRX* high vs. low group. (**c**) A forest plot represents the top 10 pathways that are positively and negatively enriched pathways using the NES score. NOM, normalized *p*-value.

**Figure 7 cancers-12-02082-f007:**
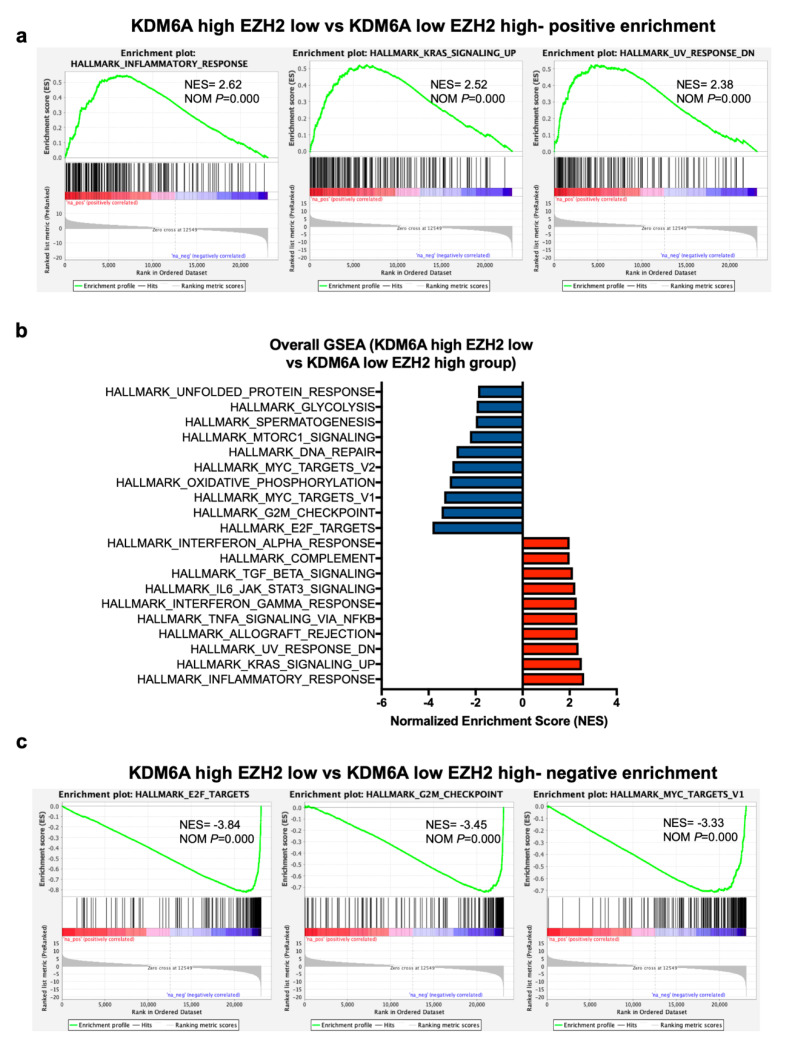
Gene set enrichment analysis of *KDM6A* high-*EZH2* low and *KDM6A* low-*EZH2* high group. (**a**) Top 3 enrichment plots showing the positively and (**b**) negatively enriched pathways in *KDM6A* high-*EZH2* low vs. *KDM6A* low-*EZH2* high group. (**c**) A forest plot represents the top 10 pathways that are positively and negatively enriched pathways using the NES score. NOM, normalized *p*-value.

**Table 1 cancers-12-02082-t001:** (a) Effect of gene-gene and gene-sex interaction on overall survival (OS) in SKCM cohort. Numbers representing the *p*-value where *p* < 0.05 refers to significant interaction on OS. (b) Effect of interaction on MSS in LMC (Validation cohort).

(a) Effect of gene-gene and gene-sex interaction on OS in SKCM cohort				
Overall	*KDM6A*	*ATRX*	*EZH2*	Sex
*KDM6A*	-	0.983	**0.033**	0.96
*ATRX*		-	0.313	0.14
*EZH2*			-	**0.02**
Sex				-
Primary				
*KDM6A*	-	0.68	0.32	0.734
*ATRX*		-	0.878	0.971
*EZH2*			-	**0.045**
Sex				-
Metastases				
*KDM6A*	-	0.991	**0.042**	0.898
*ATRX*		-	0.25	0.115
*EZH2*			-	**0.049**
Sex				-
(b) Effect of interaction on MSS in LMC (Validation cohort).				
Primary	*KDM6A*	*ATRX*	*EZH2*	Sex
*KDM6A*	-	**0.05**	**0.03**	0.41
*ATRX*		-	0.10	0.09
*EZH2*			-	**0.02**
Sex				**-**

Bold *p*-value refers to significance where, *p* < 0.05. Gene names are represented as italic.

**Table 2 cancers-12-02082-t002:** Correlation between immune cell subsets and *KDM6A, ATRX* based on gender.

Genes	CD4+ T Cells		CD8+ T Cells		B Cells		DC		Neutrophil		Macrophage	
	Pearson	*p*	Pearson	*p*	Pearson	*p*	Pearson	*p*	Pearson	*p*	Pearson	*p*
	r		r		r		r		r		r	
*KDM6A* Overall (*N* = 458)	**0.229**	**<0.0001**	**0.32**	**<0.0001**	**0.18**	**0.0001**	**0.16**	**0.0005**	**0.31**	**<0.0001**	**0.12**	**0.008**
*KDM6A* Female (*N* = 162)	**0.24**	**0.0002**	**0.37**	**<0.0001**	**0.31**	**<0.0001**	**0.28**	**0.0002**	**0.47**	**<0.0001**	**0.24**	**0.002**
*KDM6A* Male (*N* = 274)	0.07	0.2	0.12	0.03	0.07	0.24	0.03	0.57	**0.2**	**0.008**	0.1	0.08
*KDM6A* Metastases (*N* = 355)	**0.138**	**0.0096**	**0.24**	**<0.0001**	**0.18**	**0.0006**	**0.152**	**0.004**	**0.3**	**<0.0001**	0.08	0.12
*KDM6A* Female (*N* = 127)	**0.2**	**0.019**	**0.4**	**<0.0001**	**0.33**	**0.0001**	**0.28**	**0.0014**	**0.45**	**<0.0001**	**0.238**	**0.0074**
*KDM6A* Male (*N* = 218)	0.06	0.32	**0.15**	**0.022**	0.05	0.43	0.03	0.56	**0.23**	**0.0005**	0.08	0.2
*ATRX* Overall (*N* = 458)	**0.12**	**0.006**	**0.18**	**<0.0001**	**0.12**	**0.0007**	**0.11**	**0.01**	**0.29**	**<0.0001**	**0.17**	**0.0002**
*ATRX* Female (*N* = 162)	**0.16**	**0.03**	**0.24**	**0.0002**	**0.19**	**0.01**	**0.2**	**0.0095**	**0.38**	**<0.0001**	**0.23**	**0.0002**
*ATRX* Male (*N* = 274)	0.07	0.19	**0.13**	**0.03**	0.06	0.3	0.03	0.56	**0.21**	**0.0004**	**0.14**	**0.01**
*ATRX* Metastases (*N* = 355)	**0.1669**	**0.002**	0.0643	0.2665	0.1027	0.06	0.0586	0.2808	**0.2315**	**<0.0001**	0.0852	0.2218
*ATRX* Female (*N* = 127)	**0.2217**	**0.04**	-0.0197	0.9216	0.1657	0.1344	0.02926	0.7929	**0.2268**	**0.04**	0.1704	0.1235
*ATRX* Male (*N* = 218)	0.1239	0.07	0.06161	0.3814	0.0524	0.4521	0.02607	0.7113	**0.2115**	**0.0024**	0.06546	0.4357
*KMD6B* Overall (*N* = 458)	**0.15**	**0.0007**	**−0.18**	**<0.0001**	−0.014	0.17	-0.051	0.28	**−0.11**	**0.019**	0.071	0.12
*EZH2* Overall (*N* = 458)	−0.06	0.19	**0.183**	**0.0013**	0.084	0.07	0.069	0.14	**0.18**	**<0.0001**	−0.01	0.71

Bold Pearson r value refers to significance correlation where *p* < 0.05. Gene names are represented as italic.

**Table 3 cancers-12-02082-t003:** Correlation between immune markers and *KDM6A* based on gender.

Gene	CD141		BATF3		CD103		CD25		CD69		CCR5		CXCR3		IFNG	
	(DC)		(DC)		(DC)		(Treg)		Trem		CD8		CD4			
	Spearman	*p*	Spearman	*p*	Spearman	*p*	Spearman	*p*	Spearman	*p*	Spearman	*p*	Spearman	*p*	Spearman	*p*
	r		r		r		r		r		r		r		r	
*KDM6A* Overall (*N* = 458)	−0.03	0.51	−0.08	0.07	0.01	0.73	**0.3**	**<0.0001**	**0.32**	**<0.0001**	**0.18**	**0.0001**	0.01	0.73	**0.15**	**0.0008**
*KDM6A* Female (*N* = 162)	−0.03	0.66	−0.02	0.74	0.05	0.47	**0.41**	**<0.0001**	**0.46**	**<0.0001**	**0.33**	**<0.0001**	**0.17**	**0.02**	**0.28**	**0.0003**
*KDM6A* Male (*N* = 274)	−0.06	0.26	**−0.14**	**0.018**	−0.04	0.43	**0.16**	**0.0007**	**0.21**	**0.0006**	0.04	0.49	**−0.13**	**0.03**	0.05	0.41

Bold Pearson r value refers to significance correlation where *p* < 0.05. Gene names are represented as italic.

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
