# Peer review of "Study of the Female Sex Survival Advantage in Melanoma—A Focus on X-Linked Epigenetic Regulators and Immune Responses in Two Cohorts"

_cancers, 2020, doi:10.3390/cancers12082082_

Round 1

Reviewer 1 Report

Over all the manuscript is OK.  In Fig. 1 I am a bit confused over the stratification of low vs. high for all and for females and males.   In panels b and c were the male and females KDM6A data pooled to determine an over all median for stratification or was the data divided based on results shown in panel d with a different median for males and females? 

Author Response

R: Over all the manuscript is OK.  In Fig. 1 I am a bit confused over the stratification of low vs. high for all and for females and males.   In panels b and c were the male and females KDM6A data pooled to determine an over all median for stratification or was the data divided based on results shown in panel d with a different median for males and females? 

A: We would like to thank the reviewer for the comments. Figure 1b and c refer to the overall survival of all patients according to the expression of KDM6A and KDM6B stratified above and below the median expression (>median=high, <median=low). However, in Figure 1e and f we have stratified the SKCM patients into males and females and then examined their survival according to the expression of KDM6A and KDM6B above and below the median expression.

Reviewer 2 Report

The authors analyzed KDM6A, ATRX, EZH2 and et al. genes expression level in melanoma patient in TCGA and show high KDM6A showed strong associations with immune responses and down regulation of genes associated with oncogenic pathways. Overall, it is an interesting study.

Line 106, KDM6A mRNA expression was associated with better overall survival in women, but not men. This conclusion was from Figure 1e and Figure 1f. And the association of KDM6A mRNA expression and overall survival of SKCM is not impressive from your Figure 1b.

Minor point: And it is better to keep high expression in one color and low expression in another color in your figure 1b and 1c, 1e and 1f.

Line 139, High ATRX was associated with improved survival regardless of KDM6A expression level and vice-verse, in your figure 2b, when KDM6A is high, your conclusion is correct, however, when KDM6A is low, it seems ATRX low has a slightly better survival.

Line 159-line 167, each panel of the supplemental figures should have figure legend.

Line 231, how do you define HALLMARK ALLOGRAFT REJECTION, INFLAMMATORY RESPONSE and INTERFERON GAMMA RESPONSE? what genes are included, you didn’t cite any paper or any information.

It is unclear that how you make Figure 4 to Figure 6, materials and methods are too brief to get enough information. And in figure 1, KDM6A mRNA expression was associated with better overall survival in women, but not men. For figure 4, is it possible to look at gene enrichment in both men and women patients. Will women have better immune response?

Author Response

R: The authors analyzed KDM6A, ATRX, EZH2 and et al. genes expression level in melanoma patient in TCGA and show high KDM6A showed strong associations with immune responses and down regulation of genes associated with oncogenic pathways. Overall, it is an interesting study.

Line 106, KDM6A mRNA expression was associated with better overall survival in women, but not men. This conclusion was from Figure 1e and Figure 1f. And the association of KDM6A mRNA expression and overall survival of SKCM is not impressive from your Figure 1b.

A: Apologies this was a typographical error in the text. Figure 1b represents the overall survival related to KDM6A while Figure 1e and f show the overall survival related to the sex of the patients. Corrected accordingly in line 108.

R: Minor point: And it is better to keep high expression in one color and low expression in another color in your figure 1b and 1c, 1e and 1f.

A: Changed accordingly with consistent color for high or low expression in figure 1b, c, e and f.

R: Line 139, High ATRX was associated with improved survival regardless of KDM6A expression level and vice-verse, in your figure 2b, when KDM6A is high, your conclusion is correct, however, when KDM6A is low, it seems ATRX low has a slightly better survival.

A: We agree; however, KDM6A low and ATRX low group was not significant. Hence, we have also performed a multivariate cox regression analysis of KDM6A and ATRX which suggests they are independent prognostic factors.

R: Line 159-line 167, each panel of the supplemental figures should have figure legend.

A: Figure legends are included for each panel of the supplementary figures. Please see the supplementary files.

R: Line 231, how do you define HALLMARK ALLOGRAFT REJECTION, INFLAMMATORY RESPONSE and INTERFERON GAMMA RESPONSE? what genes are included, you didn’t cite any paper or any information.

A: Gene lists are included as a supplementary file in the excel sheet. GSEA was discussed in the discussion section. Additional discussion is now added in line 436-453.

R: It is unclear that how you make Figure 4 to Figure 6, materials and methods are too brief to get enough information. And in figure 1, KDM6A mRNA expression was associated with better overall survival in women, but not men. For figure 4, is it possible to look at gene enrichment in both men and women patients. Will women have better immune response?

A: A detailed methodology of GSEA is now included in 506-509 and 516-524. Additional analysis on GSEA of KDM6A has been done based on the sex of the patients. The results show that females with high KDM6A showed positive enrichment of INTERFERON GAMMA RESPONSE whereas there was no enrichment of immune-related pathways in male patients. The new GSEA results are now included as figure 5, therefore, figure 5 and 6 renumbered as figure 6 and 7. The description is provided in line 270-279 and 413-416.

Reviewer 3 Report

The paper “Study of the female sex survival advantage in melanoma – a focus on X-linked epigenetic regulators and immune responses in two cohorts” by Emran et. al presented an interesting idea and analysis of two independent melanoma patient datasets to look at potential X-linked genes in predicting overall survival in melanoma. The paper overall is well written, and the experiments carefully planned. I would suggest a few more improvements and clarification before this paper could be published. Figure 1b might be mislabeled or the text under results 2.1 might be mis-written, as it does not show high KDM6A expression is correlated with better survival in women but not men, which is actually showed in 1e and 1f. In addition, since KDM6A expression level is higher in female compared to male patients, the KDM6A high group in men might have lower KDM6A level in KDM6A high group in women. This difference might make the comparison of survival less meaningful between two sexes, especially is Fig.1b is showing when combined the two sexes, the KDM6A level does not correlate with patient survival. Can the authors explain why? In addition, the authors should indicate the disease status of the patients studied. Patients with early stage disease should not be mixed with patients with later stage, ie. Metastatic disease when analyzing survival, or they should be analyzed separately. Does EZH2 expression level differ between female and male patients? Does it correlate or anti-correlate with KDM6A levels? The authors did not explain how they calculate the interaction between two genes in the tables following Fig.1 (no table title, no legend and no description in method section). If ATRX is not correlated as showed in Table 1 (again not labeled), what is the rational to analyze their interaction in predicting overall survival in Fig.2? In Fig.S2, the P value between each group should be labeled in the Kaplan-Meier curves. According to Fig.S2, high ATRX seemed to correlate with better survival in male (if comparing yellow vs. red line). The legends should be showed separately on the side of the graph for clearer visualization. And instead of combining the plots for male and female, they should be shown separately as there’s no need to compare between high/low ATRX group in male and female (same in Fig.S3 for EZH2). In the paper it mentioned the improved survival in women is confined to the cohort with age below 60. Does age play a role in the improved survival in KDM6A high and EZH2 low group as well? If age is an important factor, KDM6A levels and its correlation with survival should also be looked at in different age groups of melanoma patients. The authors also looked at KDM6A level in several female organs regardless of age and compared to male, which does not completely make sense in what they are trying to show from this analysis. Recent studies in melanoma mouse models have shown that the Baft3+/CD103+ DCs are involved in anti-tumor immune response. The authors should also check and see if any of the genes in interest is correlated with this special population of DCs. If tumor samples collected from patients are still available, the authors could include some slides and stain for immune cell markers to validate immune cell infiltration associated with KDM6A, ATRX or EZH2.

Author Response

R: The paper “Study of the female sex survival advantage in melanoma – a focus on X-linked epigenetic regulators and immune responses in two cohorts” by Emran et. al presented an interesting idea and analysis of two independent melanoma patient datasets to look at potential X-linked genes in predicting overall survival in melanoma. The paper overall is well written, and the experiments carefully planned. I would suggest a few more improvements and clarification before this paper could be published.

Figure 1b might be mislabeled or the text under results 2.1 might be mis-written, as it does not show high KDM6A expression is correlated with better survival in women but not men, which is actually showed in 1e and 1f.

A: Apologies for the typographical error which has been corrected in line 108.

R: In addition, since KDM6A expression level is higher in female compared to male patients, the KDM6A high group in men might have lower KDM6A level in KDM6A high group in women. This difference might make the comparison of survival less meaningful between two sexes, especially is Fig.1b is showing when combined the two sexes, the KDM6A level does not correlate with patient survival. Can the authors explain why?

A: Since KDM6A is located on the X-chromosome, therefore, both alleles transcribed in women which led to higher expression in women compared to men. This resulted in a better survival advantage in women melanoma patients. However, Log rank p-value of combined male and female patients is 0.033, which is slightly lower than the female group (0.042). This may indicate that .some male patients with high KDM6A level might have improved survival.

R: In addition, the authors should indicate the disease status of the patients studied. Patients with early stage disease should not be mixed with patients with later stage, ie. Metastatic disease when analyzing survival, or they should be analyzed separately.

A: We have analysed the survival of KDM6A, ATRX and EZH2 for primary and metastatic samples separately. However, it was not significant suggesting contribution from both disease stage in the SKCM cohort. The additional analysis included as supplementary table1 and in line 115-118 in the main text.

R: Does EZH2 expression level differ between female and male patients? Does it correlate or anti-correlate with KDM6A levels?

A: EZH2 expression is relatively higher in male compared to female, however statistically not significant. EZH2 partially correlated with KDM6A level. However, paired analysis suggests that a subset of patients with high KDM6A level had significantly low EZH2 level and vice versa. This is now included as supplementary figure 4a-c and in line 191-195.

R: The authors did not explain how they calculate the interaction between two genes in the tables following Fig.1 (no table title, no legend and no description in method section).

A: Statistical interaction details have been provided in the methodology section in line 534-537. Additional description is included in Table1.

R: If ATRX is not correlated as showed in Table 1 (again not labeled), what is the rational to analyze their interaction in predicting overall survival in Fig.2?

A: We asked whether two x-linked genes have a survival advantage in melanoma patients. The analysis also addresses the question of whether or not KDM6A and ATRX act independently to each other to modulate H3K27me3 level and hence target genes.

R: In Fig.S2, the P value between each group should be labeled in the Kaplan-Meier curves. According to Fig.S2, high ATRX seemed to correlate with better survival in male (if comparing yellow vs. red line). The legends should be showed separately on the side of the graph for clearer visualization. And instead of combining the plots for male and female, they should be shown separately as there’s no need to compare between high/low ATRX group in male and female (same in Fig.S3 for EZH2).

A: Thanks for your suggestion. It has been corrected accordingly. Please see the supplementary file.

R: In the paper it mentioned the improved survival in women is confined to the cohort with age below 60. Does age play a role in the improved survival in KDM6A high and EZH2 low group as well? If age is an important factor, KDM6A levels and its correlation with survival should also be looked at in different age groups of melanoma patients.

A: The reference to the loss of survival advantage for females over the age of 60 came from a reference we quote. This was not confirmed in our study. We have looked at age as both continuous and below and above mean variable with KDM6A high and EZH2 low group, however, it is not significant. Please see supplementary figure 4d and in line 195-199.

R: The authors also looked at KDM6A level in several female organs regardless of age and compared to male, which does not completely make sense in what they are trying to show from this analysis.

A: GTEx database does not contain the information of age of the control subject so it would not be possible to show the distribution according to that. The key focus in this figure was to show high KDM6A level in lymphocytes in female compared to male. This would be consistent with the immune-related pathway enrichment associated with high KDM6A.

R: Recent studies in melanoma mouse models have shown that the Baft3+/CD103+ DCs are involved in anti-tumor immune response. The authors should also check and see if any of the genes in interest is correlated with this special population of DCs.

A: We appreciate this comment. We have analysed this accordingly and included it in table 3. However, it is not correlated with KDM6A.

R: If tumor samples collected from patients are still available, the authors could include some slides and stain for immune cell markers to validate immune cell infiltration associated with KDM6A, ATRX or EZH2. 

A: Unfortunately, this is beyond the capacity of our current project but we aim to pursue this in future studies.

Round 2

Reviewer 2 Report

The authors have address the majority of concerns I had with the original submission. 

Reviewer 3 Report

Dear authors,

Thank you for addressing the all the review comments thoroughly. I would like to congratulate you on the nicely performed study and recommend acceptance of the paper.